# Deep Koopman Learning using Noisy Data

**Wenjian Hao**                                                    *hao93@purdue.edu*
*School of Aeronautics and Astronautics*
*Purdue University*

**Devesh Upadhyay**                                    *devesh.upadhyay@saabinc.com*
*Saab, Inc.*

**Shaoshuai Mou**                                                  *mous@purdue.com*
*School of Aeronautics and Astronautics*
*Purdue University*

**Reviewed on OpenReview:** *https://openreview.net/forum?id=j6Rm6T2lFU*

## Abstract

This paper proposes a data-driven framework to learn a finite-dimensional approximation of a Koopman operator for approximating the state evolution of a dynamical system under noisy observations. To this end, our proposed solution has two main advantages. First, the proposed method only requires the measurement noise to be bounded. Second, the proposed method modifies the existing deep Koopman operator formulations by characterizing the effect of the measurement noise on the Koopman operator learning and then mitigating it by updating the tunable parameter of the observable functions of the Koopman operator, making it easy to implement. The performance of the proposed method is demonstrated on several standard benchmarks. We then compare the presented method with similar methods proposed in the latest literature on Koopman learning.

## 1 Introduction

Directly dealing with complex nonlinear dynamical systems for model-based control design has remained a challenge for the control community. One long-standing solution to this problem has been to use linearized models and the associated vast body of knowledge for linear analysis. Linear control theory is a very rich and well-developed field that provides rigorous control development with methods for stability and robustness guarantees. Lyapunov showed that for a linearized system that is stable around an equilibrium point, there exists a region of stability around this equilibrium point for which the original nonlinear system is also stable A.Lyapunov (1992). Recent advances in data-driven methods have spurred new and increased research interest in machine learning (ML) based methods for deriving reduced-order models (ROM) as surrogates for complex nonlinear systems. This has also led to the adoption of these methods for developing control and autonomy/automation solutions for robotic and unmanned systems. Examples include learning dynamics using deep neural networks (DNNs) Murphy (2002); Gillespie et al. (2018), physics-informed neural networks (PINNs) Raissi et al. (2019), and lifting linearization methods such as Koopman operator methods Koopman (1931); Koopman & Neumann (1932); Mezić (2015); Proctor et al. (2018); Mauroy & Goncalves (2016).

Lifting linearization allows one to represent a nonlinear system with an equivalent linear system in a lifted, higher-dimensional space. One widely adopted lifting method is the extending dynamic mode decomposition (EDMD) Williams et al. (2015), which lifts the state space to a higher-dimensional space, for which the temporal evolution is approximately linear Korda & Mezić (2018). It is, however, typically difficult to find an exact finite-dimensional linear representation for most nonlinear systems. Further, the Koopman operator focuses on non-autonomous systems: $\frac{d\boldsymbol{x}}{dt} = \boldsymbol{f}(\boldsymbol{x}, \boldsymbol{u}, t)$. This poses a challenge for the control design of dynamical systems in the choice of a sufficient basis function necessary for the lifted system to be linear

and exact. Extensions to such systems require truncation-based approximations, and the finite-dimensional representation is no longer exact. To this end, various eigen-decomposition-based truncations are proposed. In Lusch et al. (2017) the authors proposed using deep learning methods to discover the eigenfunctions of the approximated Koopman operator, and Yeung et al. (2019); Lusch et al. (2018); Han et al. (2020); Bevanda et al. (2021) employed DNNs as observable functions of the Koopman operator, which are tuned based on collected state-control pairs by minimizing an appropriately defined loss function which is also referred as the deep Koopman operator method (DKO). Recent work, such as Hao et al. (2024), has extended the DKO method to approximate nonlinear time-varying systems. These methods rely on a set of measured output variables that collectively define some nonlinear representation of the independent state variables. Establishing a sufficient set of these observable functions remains an active area of research. Further, real-world noisy measurements impose additional challenges. Additionally, for most practical systems, it is also critical to find computationally feasible approximation methods for extracting finite-dimensional representations.

**Related work.** While Koopman-based methods have been proven to be effective in learning dynamics from a system's input-output data pairs. However, in practical real-world applications, the output measurements are noisy and can result in biased estimates of the linear system. Even if the noise of the state variables is assumed to be uncorrelated, the nonlinear transformations in the observables may lead to complex noise-influence correlations between the noise-free states and the transformed observables. Several methods are proposed to solve the measurement noise issue. One solution Sotiropoulos. (2021) is to directly measure the states and the observables, this, however, may not always be possible. Noisy measurements are also shown to further complicate the anti-causal observable problem when dealing with the lifting of controlled systems Selby (2021). In other approaches, authors in Dawson et al. (2016); Hemati et al. (2017) introduce total least square (TLS) methods in DMD, in Haseli & Cortés (2019) the authors propose a combination of the EDMD and TLS methods to account for the measurement noise, and in Sinha et al. (2020); Wanner & Mezic (2022); Sinha et al. (2023) the authors propose to solve the EDMD with measurement noise as a robust Koopman operator problem which is a min-max optimization problem.

This paper extends the DKO method to the scenario where the system state data is corrupted by unknown but bounded measurement noise. As already discussed, this creates the challenge of generating additional noise transformations impacted by the DNN-derived basis functions of the deep Koopman operator. This leads to distortion of the noise, and the properties of the measurement noise and associated correlations may not remain the same after lifting. The contributions of this work are that we first propose a data-driven framework to learn the deep Koopman operator from the system states-inputs data pairs under unknown and bounded measurement noise, and then we provide numerical evidence that our proposed method can approximate the system dynamics with reasonable accuracy adequate for control applications.

This paper is organized as follows. Section 2 states the problem. Section 3 presents the proposed algorithm and its theoretical development. The numerical simulations and comparison of the algorithms are shown in Section 4. Finally, Section 5 concludes the paper.

**Notations**. We denote $\| \cdot \|$ as the Euclidean norm. For a matrix $\boldsymbol{A} \in \mathbb{R}^{m \times n}$, $\| \boldsymbol{A} \|_F$ denotes its Frobenius norm, $\boldsymbol{A}'$ denotes its transpose, and $\boldsymbol{A}^{\dagger}$ denotes its Moore-Penrose pseudoinverse. Additionally, $\mathbf{I}_n$ represents the $n \times n$ identity matrix.

## 2 Problem

Consider the following time-invariant system:

$$\boldsymbol{x}(t+1) = \boldsymbol{f}(\boldsymbol{x}(t), \boldsymbol{u}(t)), \quad \boldsymbol{x}(0) \text{ given}, \tag{1}$$

$$\boldsymbol{y}(t) = \boldsymbol{x}(t) + \boldsymbol{w}(t), \tag{2}$$

where $t = 0, 1, 2, \cdots$ denotes the time index, $\boldsymbol{x}(t) \in \mathbb{R}^n$ and $\boldsymbol{u}(t) \in \mathbb{R}^m$ denote the system state and control input, respectively, $\boldsymbol{y}(t) \in \mathbb{R}^n$ denotes the measured state, $\boldsymbol{w}(t) \in \mathbb{R}^n$ corresponds to the unknown measurement noise, which is assumed to be bounded (i.e., $\| \boldsymbol{w}(t) \| \leq w_{max}$), and $\boldsymbol{f} : \mathbb{R}^n \times \mathbb{R}^m \to \mathbb{R}^n$ denotes the dynamics mapping which is assumed to be unknown.

Suppose an observed system states-inputs trajectory from time $0$ to $T$ is denoted as:

$$\boldsymbol{\xi} = \{(\boldsymbol{y}_t, \boldsymbol{u}_t), t = 0, 1, 2, \cdots, T\}. \tag{3}$$

One approach to approximating the unknown dynamics $\boldsymbol{f}$ in Eq. (1) using $\boldsymbol{\xi}$ is through the deep Koopman operator (DKO) method. We begin with an approximate representation of the original system dynamics as $\hat{\boldsymbol{x}}(t + 1) = \hat{\boldsymbol{f}}(\hat{\boldsymbol{x}}(t), \boldsymbol{u}(t), \boldsymbol{\theta})$ with $\hat{\boldsymbol{x}}(0) = \boldsymbol{x}(0)$, where $\hat{\boldsymbol{x}}(t) \in \mathbb{R}^n$ represents the system state. The function $\hat{\boldsymbol{f}}$ is constructed based on the Koopman operator theory-based process, as described by:

$$\boldsymbol{g}(\hat{\boldsymbol{x}}(t+1), \boldsymbol{\theta}) = \boldsymbol{A}\boldsymbol{g}(\hat{\boldsymbol{x}}(t), \boldsymbol{\theta}) + \boldsymbol{B}\boldsymbol{u}(t), \tag{4}$$

$$\hat{\boldsymbol{x}}(t) = \boldsymbol{C}\boldsymbol{g}(\hat{\boldsymbol{x}}(t), \boldsymbol{\theta}), \tag{5}$$

where $\boldsymbol{g}(\cdot, \boldsymbol{\theta}) : \mathbb{R}^n \to \mathbb{R}^r$ is represented by a Lipschitz continuous DNN with a known architecture and tunable parameters $\boldsymbol{\theta} \in \mathbb{R}^p$, and $\boldsymbol{A} \in \mathbb{R}^{r \times r}$, $\boldsymbol{B} \in \mathbb{R}^{r \times m}$, $\boldsymbol{C} \in \mathbb{R}^{n \times r}$ are variable matrices to be determined later. Here, Eq. (4) with $r \geq n$ represents the dynamics evolution in the lifted space $\mathbb{R}^r$, and Eq. (5) defines the mapping between the lifted space $\mathbb{R}^r$ and the original space $\mathbb{R}^n$. By integrating Eq. (4)-(5), the deep Koopman operator dynamics can be expressed as follows:

$$\hat{\boldsymbol{x}}(t+1) = \hat{\boldsymbol{f}}(\hat{\boldsymbol{x}}(t), \boldsymbol{u}(t), \boldsymbol{\theta}) = \boldsymbol{C}(\boldsymbol{A}\boldsymbol{g}(\hat{\boldsymbol{x}}(t), \boldsymbol{\theta}) + \boldsymbol{B}\boldsymbol{u}(t)), \quad \hat{\boldsymbol{x}}(0) = \boldsymbol{x}(0). \tag{6}$$

The **problem of interest** is to determine the constant matrices $\boldsymbol{A}^*$, $\boldsymbol{B}^*$, $\boldsymbol{C}^*$ and the optimal parameter $\boldsymbol{\theta}^*$ using the noisy trajectory $\boldsymbol{\xi}$ in Eq. (3) such that, for any $0 \leq t \leq T - 1$, the following approximation holds:

$$\boldsymbol{g}(\boldsymbol{y}_{t+1}, \boldsymbol{\theta}^*) = \boldsymbol{A}^* \boldsymbol{g}(\boldsymbol{y}_t, \boldsymbol{\theta}^*) + \boldsymbol{B}^* \boldsymbol{u}_t, \tag{7}$$

$$\boldsymbol{x}_t = \boldsymbol{C}^* \boldsymbol{g}(\boldsymbol{y}_t, \boldsymbol{\theta}^*). \tag{8}$$

For notational brevity, we define the set of $\boldsymbol{A}^*$, $\boldsymbol{B}^*$, $\boldsymbol{C}^*$, $\boldsymbol{\theta}^*$ satisfying Eq. (7)-(8) as the Deep Koopman Representation (DKR), which will be referenced throughout this paper.

$$\mathcal{K} = \{\boldsymbol{A}^*, \boldsymbol{B}^*, \boldsymbol{C}^*, \boldsymbol{\theta}^*\}. \tag{9}$$

## 3 Main Results

This section first outlines the main challenges and key ideas underlying the proposed approach, followed by the presentation of an algorithm to achieve the DKR in Eq. (9).

### 3.1 Challenges and Key Ideas

To achieve the DKR, one natural approach is to minimize the following estimation errors using $\boldsymbol{\xi}$ in Eq. (3):

$$\boldsymbol{A}^*, \boldsymbol{B}^*, \boldsymbol{C}^*, \boldsymbol{\theta}^* = \arg \min_{\boldsymbol{A}, \boldsymbol{B}, \boldsymbol{C}, \boldsymbol{\theta}} \frac{1}{2T} \sum_{t=0}^{T-1} \|\boldsymbol{x}_{t+1} - \hat{\boldsymbol{f}}(\boldsymbol{y}_t, \boldsymbol{u}_t, \boldsymbol{\theta})\|^2. \tag{10}$$

A fundamental challenge in solving Eq. (10) arises from the fact that the true system states $\boldsymbol{x}_t$ are unknown in our problem setting.

To address this challenge, we propose an alternative minimization problem to Eq. (10). To proceed, for any $0 \leq t \leq T - 1$, we first introduce the notation:

$$\boldsymbol{x}_{t+1} = \tilde{\boldsymbol{f}}(\boldsymbol{x}_t, \boldsymbol{u}_t, \tilde{\boldsymbol{\theta}}^*) + \tilde{\boldsymbol{\epsilon}}_t = \tilde{\boldsymbol{C}}^*(\tilde{\boldsymbol{A}}^* \boldsymbol{g}(\boldsymbol{x}_t, \tilde{\boldsymbol{\theta}}^*) + \tilde{\boldsymbol{B}}^* \boldsymbol{u}_t) + \tilde{\boldsymbol{\epsilon}}_t$$

and

$$\boldsymbol{y}_{t+1} = \bar{\boldsymbol{f}}(\boldsymbol{y}_t, \boldsymbol{u}_t, \bar{\boldsymbol{\theta}}^*) + \bar{\boldsymbol{\epsilon}}_t = \bar{\boldsymbol{C}}^*(\bar{\boldsymbol{A}}^* \boldsymbol{g}(\boldsymbol{y}_t, \bar{\boldsymbol{\theta}}^*) + \bar{\boldsymbol{B}}^* \boldsymbol{u}_t) + \bar{\boldsymbol{\epsilon}}_t,$$

where $\tilde{\boldsymbol{f}}$ and $\bar{\boldsymbol{f}}$ are introduced DKO dynamics achieved using the noise-free and noisy trajectories, respectively, based on the same function $\boldsymbol{g}(\cdot, \boldsymbol{\theta})$. The terms $\tilde{\boldsymbol{\epsilon}}_t$ and $\bar{\boldsymbol{\epsilon}}_t$ represent the estimation errors that arise from solving the following optimization problems:

$$\tilde{\boldsymbol{A}}^*, \tilde{\boldsymbol{B}}^*, \tilde{\boldsymbol{C}}^*, \tilde{\boldsymbol{\theta}}^* = \arg \min_{\boldsymbol{A}, \boldsymbol{B}, \boldsymbol{C}, \boldsymbol{\theta}} \frac{1}{2T} \sum_{t=0}^{T-1} \|\boldsymbol{x}_{t+1} - \hat{\boldsymbol{f}}(\boldsymbol{x}_t, \boldsymbol{u}_t, \boldsymbol{\theta})\|^2 \tag{11}$$

and

$$\bar{\boldsymbol{A}}^*, \bar{\boldsymbol{B}}^*, \bar{\boldsymbol{C}}^*, \bar{\boldsymbol{\theta}}^* = \arg \min_{\boldsymbol{A}, \boldsymbol{B}, \boldsymbol{C}, \boldsymbol{\theta}} \frac{1}{2T} \sum_{t=0}^{T-1} \|\boldsymbol{y}_{t+1} - \hat{\boldsymbol{f}}(\boldsymbol{y}_t, \boldsymbol{u}_t, \boldsymbol{\theta})\|^2. \tag{12}$$

Then, by expanding the error function in Eq. (10) using the introduced $\tilde{\boldsymbol{f}}$ and $\bar{\boldsymbol{f}}$, and applying the triangle inequality, we obtain:

$$\begin{aligned}
&\|\boldsymbol{x}_{t+1} - \tilde{\boldsymbol{f}}(\boldsymbol{x}_t, \boldsymbol{u}_t, \tilde{\boldsymbol{\theta}}^*) + \tilde{\boldsymbol{f}}(\boldsymbol{x}_t, \boldsymbol{u}_t, \tilde{\boldsymbol{\theta}}^*) - \bar{\boldsymbol{f}}(\boldsymbol{y}_t, \boldsymbol{u}_t, \bar{\boldsymbol{\theta}}^*) + \bar{\boldsymbol{f}}(\boldsymbol{y}_t, \boldsymbol{u}_t, \bar{\boldsymbol{\theta}}^*) - \boldsymbol{y}_{t+1} + \boldsymbol{y}_{t+1} - \hat{\boldsymbol{f}}(\boldsymbol{y}_t, \boldsymbol{u}_t, \boldsymbol{\theta})\|^2 \\
&\le \| \boldsymbol{y}_{t+1} - \hat{\boldsymbol{f}}(\boldsymbol{y}_t, \boldsymbol{u}_t, \boldsymbol{\theta})\|^2 + \|\tilde{\boldsymbol{f}}(\boldsymbol{x}_t, \boldsymbol{u}_t, \tilde{\boldsymbol{\theta}}^*) - \bar{\boldsymbol{f}}(\boldsymbol{y}_t, \boldsymbol{u}_t, \bar{\boldsymbol{\theta}}^*) \|^2 + \|\tilde{\boldsymbol{\epsilon}}_t\|^2 + \|\bar{\boldsymbol{\epsilon}}_t\|^2.
\end{aligned} \tag{13}$$

Note that $\tilde{\boldsymbol{\epsilon}}_t$ and $\bar{\boldsymbol{\epsilon}}_t$ are typically small positive constants. For a detailed analysis of these estimation errors, we refer to the existing work Hao et al. (2024). Removing the constant terms from the upper bound derived in Eq. (13), we formulate the following loss function to achieve the DKR in Eq. (9):

$$\mathbf{L}_f(\boldsymbol{A}, \boldsymbol{B}, \boldsymbol{C}, \boldsymbol{\theta}) = \frac{1}{2T} \sum_{t=0}^{T-1} (\| \boldsymbol{y}_{t+1} - \hat{\boldsymbol{f}}(\boldsymbol{y}_t, \boldsymbol{u}_t, \boldsymbol{\theta})\|^2 + \|\tilde{\boldsymbol{f}}(\boldsymbol{x}_t, \boldsymbol{u}_t, \tilde{\boldsymbol{\theta}}^*) - \bar{\boldsymbol{f}}(\boldsymbol{y}_t, \boldsymbol{u}_t, \bar{\boldsymbol{\theta}}^*) \|^2). \tag{14}$$

Using this development, we propose that instead of minimizing the residual in Eq. (10), we can minimize the upper bound of Eq. (10) as in Eq. (14). This minimization problem is split into two components. First, given a noisy trajectory $\boldsymbol{\xi}$, the goal is to determine a dynamical model that approximates the relationship between $\boldsymbol{y}_t, \boldsymbol{u}_t$ and $\boldsymbol{y}_{t+1}$ as stated in Eq. (12). The second component focuses on minimizing the norm difference between the model $\tilde{\boldsymbol{f}}(\boldsymbol{x}_t, \boldsymbol{u}_t, \tilde{\boldsymbol{\theta}}^*)$ from Eq. (11) and $\bar{\boldsymbol{f}}(\boldsymbol{y}_t, \boldsymbol{u}_t, \bar{\boldsymbol{\theta}}^*)$ from Eq. (12). The key challenge in solving Eq. (14) lies in quantifying the difference between the two models, $\tilde{\boldsymbol{f}}(\boldsymbol{x}_t, \boldsymbol{u}_t, \tilde{\boldsymbol{\theta}}^*)$ and $\bar{\boldsymbol{f}}(\boldsymbol{y}_t, \boldsymbol{u}_t, \bar{\boldsymbol{\theta}}^*)$.

**Remark 1** *Note that, in contrast to the conventional error function of $\|\boldsymbol{x}_{t+1} - \hat{\boldsymbol{f}}(\boldsymbol{y}_t, \boldsymbol{u}_t, \boldsymbol{\theta})\|^2 = \|\boldsymbol{y}_{t+1} - \boldsymbol{w}_{t+1} - \hat{\boldsymbol{f}}(\boldsymbol{y}_t, \boldsymbol{u}_t, \boldsymbol{\theta})\|^2 \le \|\boldsymbol{y}_{t+1} - \hat{\boldsymbol{f}}(\boldsymbol{y}_t, \boldsymbol{u}_t, \boldsymbol{\theta})\|^2 + \|\boldsymbol{w}_{t+1}\|^2$, the proposed loss function in Eq. (14) substitutes the term $\|\boldsymbol{w}_{t+1}\|^2$ with the discrepancy between the system dynamics, $\|\tilde{\boldsymbol{f}} - \bar{\boldsymbol{f}} \|^2$. This formulation allows for further minimization by tuning $\boldsymbol{\theta}$, thereby enhancing robustness and stability in estimation.*

## 3.2 Algorithm

We now introduce an algorithm to solve Eq. (14) utilizing the noisy data from Eq. (3). We start by addressing the first term in Eq. (14), for which we define the following loss function:

$$\mathbf{L}_{f,1}(\boldsymbol{A}, \boldsymbol{B}, \boldsymbol{C}, \boldsymbol{\theta}) = \frac{1}{2T} (\sum_{t=0}^{T-1} \|\boldsymbol{g}(\boldsymbol{y}_{t+1}, \boldsymbol{\theta}) - \boldsymbol{A}\boldsymbol{g}(\boldsymbol{y}_t, \boldsymbol{\theta}) - \boldsymbol{B}\boldsymbol{u}_t\|^2 + \|\boldsymbol{y}_t - \boldsymbol{C}\boldsymbol{g}(\boldsymbol{y}_t, \boldsymbol{\theta})\|^2), \tag{15}$$

where the first and second parts of $\mathbf{L}_{f,1}$ represent the estimation errors in the lifted space, as described in Eq. (4), and the original space, as outlined in Eq. (5), respectively.

If the matrices $\boldsymbol{A}, \boldsymbol{B}, \boldsymbol{C}$ are known, the optimal $\boldsymbol{\theta}^*$ that minimizes $\mathbf{L}_{f,1}$ can be directly obtained using the gradient descent method. However, when these matrices are unknown, an alternative iterative approach can be employed. Specifically, let $k = 0, 1, 2, \cdots$ be the iteration index, and let $\boldsymbol{\theta}_k$ represent the estimation of $\boldsymbol{\theta}^*$ at the $k$-th iteration. Initially, the relationship between the constant matrices and $\boldsymbol{\theta}_k$ is established based on the trajectory $\boldsymbol{\xi}$. Once this relationship is identified, the gradient $\nabla_{\boldsymbol{\theta}} \mathbf{L}_{f,1}$ can be computed only using $\boldsymbol{\theta}_k$, enabling the application of the gradient descent method to minimize $\mathbf{L}_{f,1}$ by iteratively updating $\boldsymbol{\theta}_k$.

To this end, we first introduce the following data matrices formed from $\boldsymbol{\xi}$:

$$\mathbf{Y} = [\boldsymbol{y}_0, \boldsymbol{y}_1, \cdots, \boldsymbol{y}_{T-1}] \in \mathbb{R}^{n \times T}, \quad \bar{\mathbf{Y}} = [\boldsymbol{y}_1, \boldsymbol{y}_2, \cdots, \boldsymbol{y}_T] \in \mathbb{R}^{n \times T}, \mathbf{U} = [\boldsymbol{u}_0, \boldsymbol{u}_1, \cdots, \boldsymbol{u}_{T-1}] \in \mathbb{R}^{m \times T},$$

$$\mathbf{G}_k = [\boldsymbol{g}(\boldsymbol{y}_0, \boldsymbol{\theta}_k), \boldsymbol{g}(\boldsymbol{y}_1, \boldsymbol{\theta}_k), \cdots, \boldsymbol{g}(\boldsymbol{y}_{T-1}, \boldsymbol{\theta}_k)] \in \mathbb{R}^{r \times T}, \bar{\mathbf{G}}_k = [\boldsymbol{g}(\boldsymbol{y}_1, \boldsymbol{\theta}_k), \boldsymbol{g}(\boldsymbol{y}_2, \boldsymbol{\theta}_k), \cdots, \boldsymbol{g}(\boldsymbol{y}_T, \boldsymbol{\theta}_k)] \in \mathbb{R}^{r \times T}. \tag{16}$$

It leads to the following compact form of Eq. (15) using given $\boldsymbol{\theta}_k$ and the definition of the Frobenius norm:

$$\hat{\mathbf{L}}_{f,1}(\boldsymbol{A}, \boldsymbol{B}, \boldsymbol{C}) = \frac{1}{2T} (\|\bar{\mathbf{G}}_k - [\boldsymbol{A} \ \boldsymbol{B}] \begin{bmatrix} \mathbf{G}_k \\ \mathbf{U} \end{bmatrix} \|_F^2 + \|\mathbf{Y} - \boldsymbol{C}\mathbf{G}_k\|_F^2). \tag{17}$$

If the matrices $\mathbf{G}_k \in \mathbb{R}^{r \times T}$ and $\begin{bmatrix} \mathbf{G}_k \\ \mathbf{U} \end{bmatrix} \in \mathbb{R}^{(r+m) \times T}$ in Eq. (17) have full row ranks (i.e., are right-invertible), then the parameter $\boldsymbol{\theta}_k$ can be updated using the following rule:

$$[\bar{A}_k^*, \bar{B}_k^*], \bar{C}_k^* = \arg \min_{[A,B],C} \hat{\mathbf{L}}_{f,1} = \bar{\mathbf{G}}_k \begin{bmatrix} \mathbf{G}_k \\ \mathbf{U} \end{bmatrix}^\dagger, \mathbf{Y}\mathbf{G}_k^\dagger, \tag{18}$$

$$\boldsymbol{\theta}_{k+1} = \boldsymbol{\theta}_k - \alpha_k \nabla_\theta \mathbf{L}_{f,1}(\bar{A}_k^*, \bar{B}_k^*, \bar{C}_k^*, \boldsymbol{\theta}_k), \quad \boldsymbol{\theta}_0 \text{ given}, \tag{19}$$

where the step size $\alpha_k$ satisfies the standard conditions $\sum_{k=0}^\infty \alpha_k = \infty$ and $\sum_{k=0}^\infty \alpha_k^2 < \infty$, and

$$\nabla_\theta \mathbf{L}_{f,1}(\bar{A}_k^*, \bar{B}_k^*, \bar{C}_k^*, \boldsymbol{\theta}_k) = \frac{1}{T} \sum_{t=0}^{T-1} \Big( (\nabla_\theta \boldsymbol{g}(\boldsymbol{y}_{t+1}, \boldsymbol{\theta}_k) - \bar{A}_k^* \nabla_\theta \boldsymbol{g}(\boldsymbol{y}_t, \boldsymbol{\theta}_k))'(\boldsymbol{g}(\boldsymbol{y}_{t+1}, \boldsymbol{\theta}_k) - \bar{A}_k^* \boldsymbol{g}(\boldsymbol{y}_t, \boldsymbol{\theta}_k) - \bar{B}_k^* \boldsymbol{u}_t)$$
$$-(\bar{C}_k^* \nabla_\theta \boldsymbol{g}(\boldsymbol{y}_t, \boldsymbol{\theta}_k))'(\boldsymbol{y}_t - \bar{C}_k^* \boldsymbol{g}(\boldsymbol{y}_t, \boldsymbol{\theta}_k)).$$

Note that the matrices $\bar{A}_k^*$, $\bar{B}_k^*$, and $\bar{C}_k^*$ remain constant while computing $\nabla_\theta \mathbf{L}_{f,1}(\bar{A}_k^*, \bar{B}_k^*, \bar{C}_k^*, \boldsymbol{\theta}_k)$.

To minimize the second part of Eq. (14), which quantifies the discrepancy between the two models, $\tilde{\boldsymbol{f}}(\boldsymbol{x}_t, \boldsymbol{u}_t, \tilde{\boldsymbol{\theta}}^*)$ and $\bar{\boldsymbol{f}}(\boldsymbol{y}_t, \boldsymbol{u}_t, \bar{\boldsymbol{\theta}}^*)$, we observe that this difference arises due to the measurement noise $\boldsymbol{w}_t$. To systematically characterize this discrepancy under $\boldsymbol{w}_t$, we define the following loss function:

$$\mathbf{L}_{f,2}(\bar{\boldsymbol{\theta}}^*) = \frac{1}{2T} \sum_{t=0}^{T-1} \max_{\boldsymbol{w}_t} \| \tilde{\boldsymbol{f}}(\boldsymbol{x}_t, \boldsymbol{u}_t, \tilde{\boldsymbol{\theta}}^*) - \bar{\boldsymbol{f}}(\boldsymbol{y}_t, \boldsymbol{u}_t, \bar{\boldsymbol{\theta}}^*) \|^2$$
$$= \frac{1}{2T} \max_{\boldsymbol{w}_t} (\sum_{t=0}^{T-1} \| \boldsymbol{g}(\boldsymbol{x}_t, \tilde{\boldsymbol{\theta}}^*) - \boldsymbol{g}(\boldsymbol{y}_t, \bar{\boldsymbol{\theta}}^*) \|^2 + \|[\tilde{A}^*, \tilde{B}^*] - [\bar{A}^*, \bar{B}^*]\|_F^2 + \| \tilde{C}^* - \bar{C}^* \|_F^2). \tag{20}$$

Here, we assume that the matrices $\tilde{A}^*, \tilde{B}^*, \tilde{C}^*$ and $\bar{A}^*, \bar{B}^*, \bar{C}^*$ are obtained following the same procedure outlined in Eq. (18), using the same function $\boldsymbol{g}(\cdot, \boldsymbol{\theta})$, but derived from noise-free data and noisy data, respectively. Let $\bar{\mathbf{G}}$ and $\mathbf{G}$ denote the data matrices computed with an arbitrary given $\boldsymbol{\theta}$, analogous to $\bar{\mathbf{G}}_k$ and $\mathbf{G}_k$ in Eq. (16), respectively. Given that the system state $\boldsymbol{x}_t$ in Eq. (20) is unknown within our problem setting, we now introduce the following theorem to determine the optimal $\boldsymbol{\theta}^*$ that minimizes $\mathbf{L}_{f,2}(\bar{\boldsymbol{\theta}}^*)$.

**Theorem 1** *If the unknown measurement noise $\|\boldsymbol{w}_t\|$ is bounded by $w_{max}$ and $\boldsymbol{g}(\cdot, \boldsymbol{\theta})$ is a Lipschitz continuous function, then the optimal $\boldsymbol{\theta}^*$ that minimizes the following loss function will also minimize Eq. (20):*

$$\hat{\mathbf{L}}_{f,2}(\boldsymbol{\theta}) = \frac{1}{2T} \Big( \| \left( \begin{bmatrix} \mathbf{G} \\ \mathbf{U} \end{bmatrix} \begin{bmatrix} \mathbf{G} \\ \mathbf{U} \end{bmatrix}' \right)^{-1} \|_F^2 ((\| \bar{\mathbf{G}} \|_F^2 + \| \bar{\mathbf{G}} \begin{bmatrix} \mathbf{G} \\ \mathbf{U} \end{bmatrix}^\dagger \|_F^2) \| \mathbf{G} \|_F^2 + \| \bar{\mathbf{G}} \|_F^2)$$
$$+ \| (\mathbf{G}\mathbf{G}')^{-1} \|_F^2 \| \mathbf{Y}\mathbf{G}^\dagger \|_F^2 \| \mathbf{G} \|_F^2 \Big). \tag{21}$$

Proof of Theorem 1 is given in the Appendix. Based on $\hat{\mathbf{L}}_{f,1}$ in Eq. (17) and $\hat{\mathbf{L}}_{f,2}$ in Eq. (21), one have the following loss function:

$$\hat{\mathbf{L}}_f(A, B, C, \boldsymbol{\theta}) = \frac{1}{2T} (\| \bar{\mathbf{G}} - [A, B] \begin{bmatrix} \mathbf{G} \\ \mathbf{U} \end{bmatrix} \|_F^2 + \| \mathbf{Y} - C\mathbf{G} \|_F^2) + \hat{\mathbf{L}}_{f,2}(\boldsymbol{\theta}). \tag{22}$$

In summary, rather than directly minimizing the residual in Eq. (10), this paper proposes a method to determine $A^*$, $B^*$, $C^*$, and $\boldsymbol{\theta}^*$ that minimize Eq. (22), which serves as an upper bound for Eq. (10). The proposed algorithm follows an iterative process. In each iteration $k$, the algorithm first computes the matrices $[\bar{A}_k^*, \bar{B}_k^*]$ and $\bar{C}_k^*$ using Eq. (18). Then, the gradient descent method is employed to update $\boldsymbol{\theta}_k$ to find the optimal $\boldsymbol{\theta}^*$ that minimizes $\hat{\mathbf{L}}_f$ in Eq. (22). This iterative procedure is summarized in Algorithm 1, referred to as *deep Koopman learning with the noisy data* (DKND) in the rest of this paper.

---

**Algorithm 1:** Deep Koopman learning with the noisy data (DKND)

---

**Input: Y**, $\bar{\mathbf{Y}}$, **U** in Eq. (16).

**Output:** $\boldsymbol{A}^*, \boldsymbol{B}^*, \boldsymbol{C}^*, \boldsymbol{\theta}^*$.

**Initialization:** Set the learning rate sequences $\{\alpha_k\}_{k=0}^K$ and terminal accuracy $\epsilon \geq 0$, build DNN
$\quad\quad\quad\quad \boldsymbol{g}(\cdot, \boldsymbol{\theta}) : \mathbb{R}^n \to \mathbb{R}^r$ with given $\boldsymbol{\theta}_0 \in \mathbb{R}^p$.

**for** $k = 0, 1, 2, \cdots, K$ **do**

> Compute $[\bar{\boldsymbol{A}}_k^*, \bar{\boldsymbol{B}}_k^*]$ and $\bar{\boldsymbol{C}}_k^*$ by solving Eq. (18), and construct the loss function $\hat{\mathbf{L}}_f$ in Eq. (22),
> replacing its $[\boldsymbol{A}, \boldsymbol{B}]$, $\boldsymbol{C}$, and $\boldsymbol{\theta}$ with $[\bar{\boldsymbol{A}}_k^*, \bar{\boldsymbol{B}}_k^*]$, $\bar{\boldsymbol{C}}_k^*$, and $\boldsymbol{\theta}_k$, respectively.
>
> Update the $\boldsymbol{\theta}_k$ using the gradient descent: $\boldsymbol{\theta}_{k+1} = \boldsymbol{\theta}_k - \alpha_k \nabla_{\boldsymbol{\theta}} \hat{\mathbf{L}}_f(\bar{\boldsymbol{A}}_k^*, \bar{\boldsymbol{B}}_k^*, \bar{\boldsymbol{C}}_k^*, \boldsymbol{\theta}_k)$.
>
> Stop if $\hat{\mathbf{L}}_f < \epsilon$ and save the resulting $\bar{\boldsymbol{A}}_k^*$, $\bar{\boldsymbol{B}}_k^*$, $\bar{\boldsymbol{C}}_k^*$, and $\boldsymbol{\theta}_k$ as $\boldsymbol{A}^*$, $\boldsymbol{B}^*, \boldsymbol{C}^*$, and $\boldsymbol{\theta}^*$.

**end**

---

**Remark 2** *Note that by following the definition of Moore–Penrose inverse (i.e., for any $\boldsymbol{D} \in \mathbb{R}^{m \times n}$ with full row rank, $\boldsymbol{D}^\dagger = \boldsymbol{D}'(\boldsymbol{D}\boldsymbol{D}')^{-1}$), the inverse terms $(\begin{bmatrix} \mathbf{G} \\ \mathbf{U} \end{bmatrix} \begin{bmatrix} \mathbf{G} \\ \mathbf{U} \end{bmatrix}')^{-1}$ and $(\mathbf{G}\mathbf{G}')^{-1}$ may pose challenges in computing the gradient of $\hat{\mathbf{L}}_f$. One way to address this issue is to utilize the relation $\partial_\theta \boldsymbol{K}^{-1} = -\boldsymbol{K}^{-1}(\partial_\theta \boldsymbol{K})\boldsymbol{K}^{-1}$, where $\boldsymbol{K} \in \mathbb{R}^{n \times n}$. This approach enables gradient computation involving matrix inverses in a more manageable form.*

## 4 Experiments

In this subsection, we first demonstrate the performance of the proposed algorithm by analyzing the estimation errors between the predicted system states and the true noise-free states across four benchmark dynamics: one 2D simple linear discrete time-invariant dynamics:

$$\boldsymbol{x}_{t+1} = \begin{bmatrix} 0.9 & -0.1 \\ 0 & 0.8 \end{bmatrix} \boldsymbol{x}_t + \begin{bmatrix} 0 \\ 1 \end{bmatrix} \boldsymbol{u}_t, \quad \boldsymbol{x}_0 = \begin{bmatrix} 1 \\ 0 \end{bmatrix},$$

cartpole ($\boldsymbol{x}_t \in \mathbb{R}^4, \boldsymbol{u}_t \in \mathbb{R}$) and lunar lander ($\boldsymbol{x}_t \in \mathbb{R}^6, \boldsymbol{u}_t \in \mathbb{R}^2$) examples from the Openai gym Brockman et al. (2016), and one real-world example of unmanned surface vehicles ($\boldsymbol{x}_t \in \mathbb{R}^6, \boldsymbol{u}_t \in \mathbb{R}^2$), of which the details can be found in Li et al. (2024). Then, we compare the proposed algorithm with related methods.

**Experiment setup.** In this experiment, we first gather noise-free state-input pairs $\mathcal{D} = \{(\boldsymbol{x}_t, \boldsymbol{u}_t)\}_{t=0}^T$ from the aforementioned four examples, where $\boldsymbol{u}_t$ represents randomly generated control inputs drawn from a uniform distribution bounded between $-1$ and $1$. Subsequently, we introduce three types of bounded measurement noise: Gaussian noise ($\boldsymbol{w}_t^G$) with mean $\mu = 0$ and standard deviation $\sigma = 2$, Poisson distribution ($\boldsymbol{w}_t^P$) with an expected separation $\lambda = 3$, and uniform distribution ($\boldsymbol{w}_t^U$) generated from the open interval $[-1, 2)$. To ensure bounded noise, we apply a clipping procedure to the measurement noise. These noise types are added to the system states to yield noisy measurements. Specifically, we denote the noisy measurements under Gaussian noise as $\boldsymbol{y}_t^G = \boldsymbol{x}_t + \boldsymbol{w}_t^G$. The corresponding dataset, denoted $\mathcal{D}^G = \{(\boldsymbol{y}_t^G, \boldsymbol{u}_t)\}_{t=0}^T$, is used for the experiments. To facilitate training and testing, we allocate 80% of $\mathcal{D}^G$ to train DKND (denoted as $\mathcal{D}_{train}^G$), reserving the remaining 20% for testing (denoted as $\mathcal{D}_{test}^G$). For performance evaluation, we compute the root mean square deviation (RMSD) over the test dataset $\mathcal{D}_{test}^G$:

$$RMSD(\mathcal{D}_{test}^G) = \sqrt{\frac{1}{|\mathcal{D}_{test}^G|} \sum_{(\boldsymbol{y}_t, \boldsymbol{u}_t) \in \mathcal{D}_{test}^G} \|\boldsymbol{x}_{t+1} - \hat{\boldsymbol{f}}(\boldsymbol{y}_t, \boldsymbol{u}_t, \boldsymbol{\theta}^*)\|^2},$$

where $|\mathcal{D}_{test}^G|$ denote the number of data pairs $(\boldsymbol{y}_t, \boldsymbol{u}_t)$ in $\mathcal{D}_{test}^G$, and $\hat{\boldsymbol{f}}$ represents the estimated dynamics obtained from the proposed DKND method. Additionally, we compare the performance of DKND against three baseline algorithms: DK, which solves Eq. (12) using noisy measurements $\boldsymbol{y}_t$, DMDTLS from Dawson et al. (2016), and the multilayer perceptron (MLP) approach. To fairly evaluate the algorithms, we assign the above methods with the same DNN structure, training parameters (e.g., learning rate, training epochs, etc.),

and training and testing datasets. To mitigate the influence of random initialization of DNN parameters, each gradient-based method is run for 10 experimental trials. The average RMSD and their standard deviations are reported in the tables over these 10 trials.

| RMSD | Methods | 2D example | Cartpole | Lunar lander | Surface vehicle |
|---|---|---|---|---|---|
| Gaussian noise | *proposed* | 0.1963±0.0002 | 0.3705±0.0027 | 0.4007±0.0004 | 0.3059±0.0091 |
| | DK | 0.2149±0.0106 | 0.5604±0.0037 | 0.4730±0.0051 | 0.3068±0.0027 |
| | MLP | 0.2118±0.0016 | 0.3987±0.0006 | 0.4650±0.0027 | 0.2960±0.0030 |
| | DMDTLS | 0.2483 | 29.9163 | 0.6836 | 2.1779 |
| | - | $w_{max} = 0.8$ | $w_{max} = 1.0$ | $w_{max} = 1.0$ | $w_{max} = 1.0$ |
| Poisson noise | *proposed* | 0.4431±0.0018 | 0.6975±0.0025 | 0.8269±0.0029 | 0.7642±0.0031 |
| | DK | 0.4707±0.0206 | 0.7996±0.0075 | 0.8565±0.0031 | 0.7768±0.0013 |
| | MLP | 0.4644±0.0011 | 0.6808±0.0017 | 0.8329±0.0019 | 0.7727±0.0009 |
| | DMDTLS | 0.4709 | 3.7518 | 0.9268 | 2.0631 |
| | - | $w_{max} = 1.2$ | $w_{max} = 1.3$ | $w_{max} = 1.5$ | $w_{max} = 1.5$ |
| Uniform noise | *proposed* | 1.4471±0.0089 | 2.1534±0.2912 | 2.1224±0.5110 | 1.7541±0.0177 |
| | DK | 1.7493±0.1877 | 2.3839±0.1021 | 2.6577±0.0422 | 1.5712±0.0264 |
| | MLP | 2.3287±0.0236 | 4.0224±0.0035 | 4.8612±0.0037 | 1.7127±0.0248 |
| | DMDTLS | 27.2598 | 39.2297 | 286.4929 | 24.2376 |
| | - | $w_{max} = 5.3$ | $w_{max} = 7.2$ | $w_{max} = 8.2$ | $w_{max} = 1.2$ |

Table 1: Averaged RSMD over training data.

| RSMD | Methods | 2D example | Cartpole | Lunar lander | Surface vehicle |
|---|---|---|---|---|---|
| Gaussian noise | *proposed* | 0.2074±0.0008 | 0.3974±0.0076 | 0.4877±0.0185 | 0.4539±0.0661 |
| | DK | 0.2124±0.0072 | 0.6190±0.0088 | 0.8433±0.0737 | 0.5642±0.1301 |
| | MLP | 0.3721±0.0311 | 0.8757±0.0172 | 1.9348±0.1598 | 0.7048±0.0581 |
| | DMDTLS | 0.2514 | 28.3258 | 0.6551 | 2.4107 |
| | - | $w_{max} = 0.8$ | $w_{max} = 1.0$ | $w_{max} = 1.0$ | $w_{max} = 1.0$ |
| Poisson noise | *proposed* | 0.4551±0.0014 | 0.7118±0.0030 | 0.8268±0.0255 | 0.9456±0.0485 |
| | DK | 0.4784±0.0211 | 0.8281±0.0088 | 1.0857±0.1158 | 1.0846 ±0.1584 |
| | MLP | 0.4888±0.0053 | 1.0596±0.0429 | 1.8316±0.1631 | 0.9229±0.0612 |
| | DMDTLS | 0.4709 | 4.4250 | 0.8958 | 3.3943 |
| | - | $w_{max} = 1.2$ | $w_{max} = 1.3$ | $w_{max} = 1.5$ | $w_{max} = 1.5$ |
| Uniform noise | *proposed* | 1.4832±0.0117 | 2.1362±0.2796 | 2.3323±0.6968 | 2.2739±0.1600 |
| | DK | 2.0752±0.2770 | 2.3234±0.1033 | 3.0809±0.0639 | 3.9145±0.6408 |
| | MLP | 2.6835±0.0831 | 4.8319±0.0939 | 6.2894±0.1411 | 3.1910 ±0.4081 |
| | DMDTLS | 26.7608 | 40.9191 | 288.2916 | 24.6145 |
| | - | $w_{max} = 5.3$ | $w_{max} = 7.2$ | $w_{max} = 8.2$ | $w_{max} = 1.2$ |

Table 2: Averaged RSMD over testing data.

**Results analysis.** As presented in Tables. 1-2, the proposed DKND method achieves smaller average RSMD and standard deviation on testing data when compared to other methods, even as the complexity of the dynamics is increasing. Specifically, when the noise follows a uniform distribution and $\boldsymbol{w}_t$ grows larger, the gap in RSMD between the proposed DKND and DK methods becomes more pronounced. Note that the RSMD for all gradient-based comparison methods over the training data does not show significant differences. This can be attributed to the fact that their training processes are terminated at the same terminal accuracy. Figs. 1-4 display detailed estimation error plots across the testing data, with shaded regions indicating the variability across 10 trials. Due to space limitations, additional experimental details, such as the generation of measurement noise, the structure of the DNNs, and the training parameters, are provided in the Appendix.

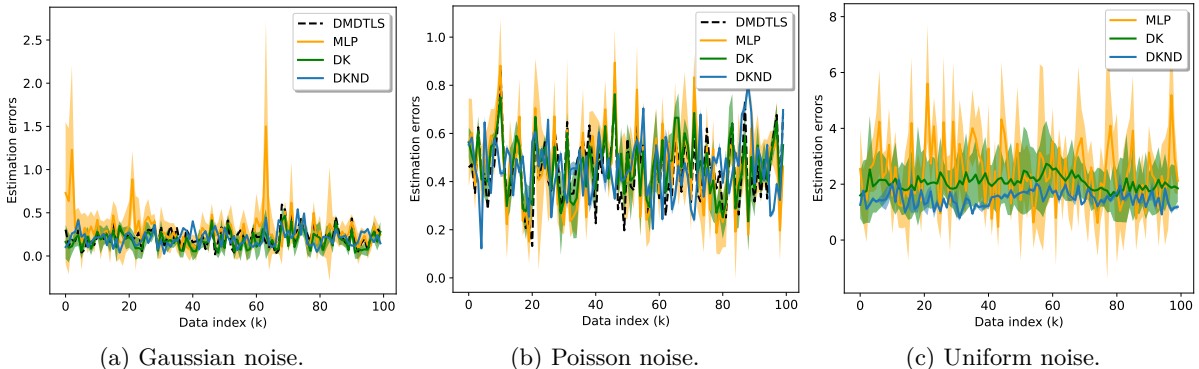

(a) Gaussian noise.      (b) Poisson noise.      (c) Uniform noise.

Figure 1: Prediction errors over testing data for the linear dynamics example.

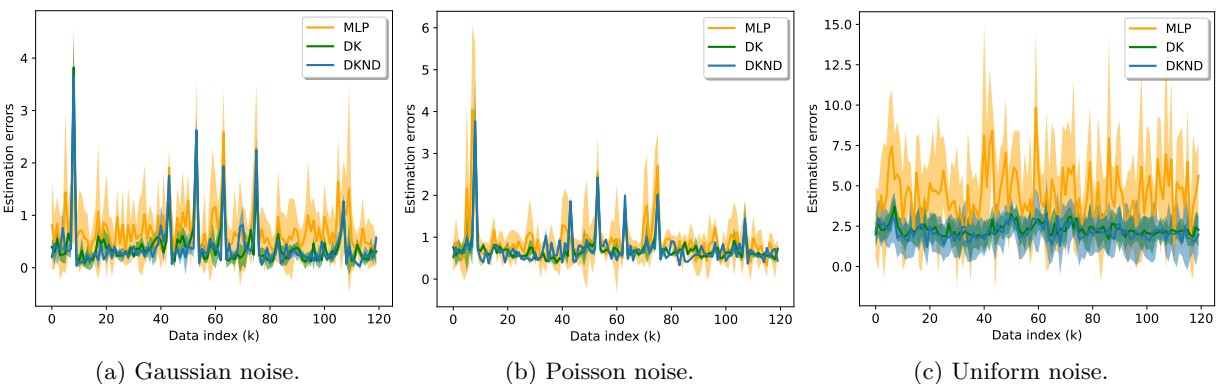

(a) Gaussian noise.      (b) Poisson noise.      (c) Uniform noise.

Figure 2: Prediction errors over testing data for the cartpole example.

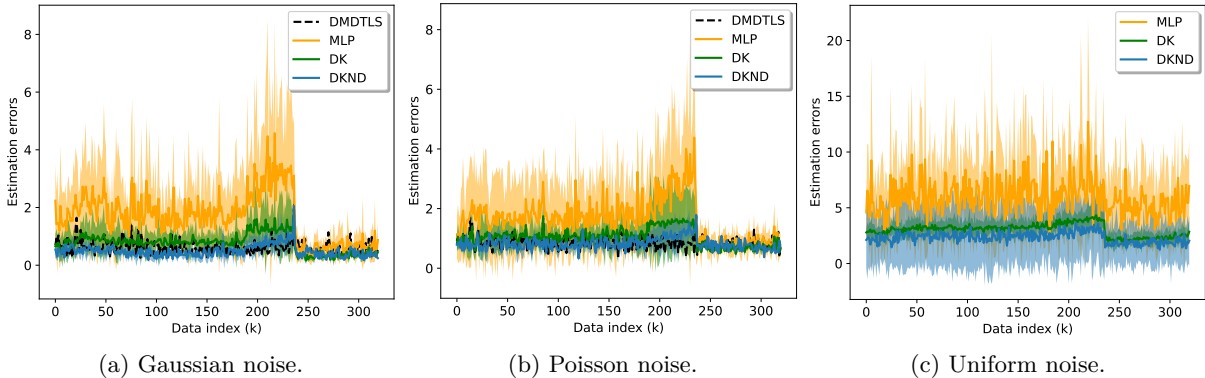

(a) Gaussian noise.      (b) Poisson noise.      (c) Uniform noise.

Figure 3: Prediction errors over testing data for the lunar lander example.

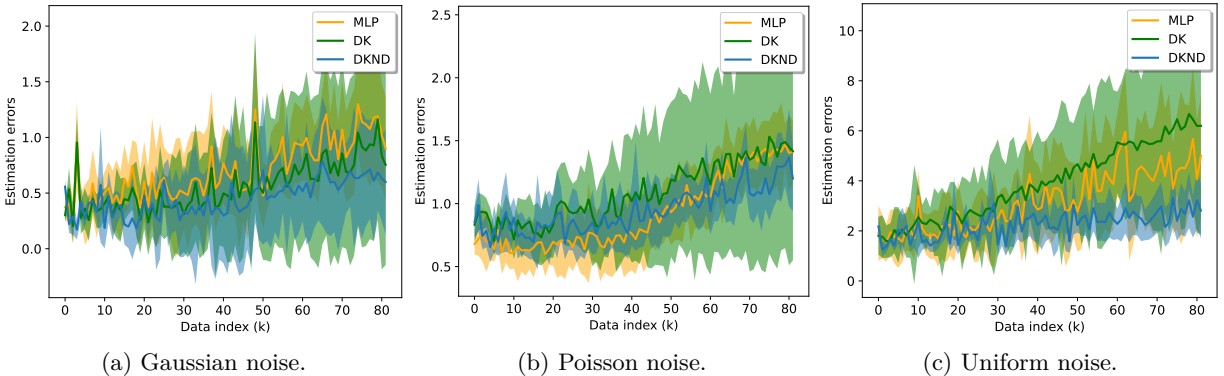

Figure 4: Prediction errors over testing data for the surface vehicle example.

# 5 Concluding Remarks

In this paper, we have introduced a data-driven framework called Deep Koopman Learning with Noisy Data (DKND) to address the challenge of learning system dynamics from data affected by measurement noise. By learning dynamics, we refer to estimating dynamics where, given $\boldsymbol{y}_t, \boldsymbol{u}_t$, the output of the estimated dynamics, $\hat{\boldsymbol{x}}_{t+1}$, approximates the true system state $\boldsymbol{x}_{t+1}$ with reasonable accuracy. The key contribution of this work lies in modifying the existing deep Koopman framework by explicitly characterizing the noise effect on the learned representation in Eq. (9) and mitigating the impact of noise on the DKR through tuning the DNN parameters to minimize Eq. (21) requiring only that the measurement noise be bounded. We evaluated the proposed DKND framework on datasets with three different types of measurement noise, using examples including simple 2D dynamics, cartpole, lunar lander, and surface vehicle systems. Our results demonstrate the robustness of DKND under different types of measurement noise compared to related methods.

**Limitations.** Since the formulation presented in this paper only addresses the scenario where the measurement noise is bounded, the effect of this bound on the performance of the proposed approach is not formally investigated and remains an open question. Due to the non-convex nature of DNN optimization, the DKND framework is inherently limited to achieving local minima. Future research could explore several aspects, including the design of optimal control strategies based on the learned dynamics using the measured noisy system states.

### Acknowledgments

This material is based upon work supported by the Defense Advanced Research Projects Agency (DARPA) via Contract No. N65236-23-C-8012 and under subcontract to Saab, Inc. as part of the RefleXAI project. Any opinions, findings and conclusions, or recommendations expressed in this material are those of the author(s) and do not necessarily reflect the views of the DARPA, the U.S. Government, or Saab, Inc.

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

## A  Appendix

In this section, we first provide the proof of Theorem 1, followed by an overview of the detailed simulation setup used in the experiments presented throughout this paper.

### A.1  Proof of Theorem 1.

Recall that $\bar{\mathbf{G}}$ and $\mathbf{G}$ denote the data matrices constructed using an arbitrary given parameter vector $\boldsymbol{\theta}$, in the same structure as $\bar{\mathbf{G}}_k$ and $\mathbf{G}_k$ in Eq. (16), respectively. Before presenting the proof, we introduce the following notions. Define $\delta \boldsymbol{g}_t = \boldsymbol{g}(\boldsymbol{y}_t, \boldsymbol{\theta}) - \boldsymbol{g}(\boldsymbol{x}_t, \boldsymbol{\theta})$ as the difference between the DNN $\boldsymbol{g}(\cdot, \boldsymbol{\theta})$ evaluated at the observed noisy system state $\boldsymbol{y}_t = \boldsymbol{x}_t + \boldsymbol{w}_t$ and the true system state $\boldsymbol{x}_t$. Next, we introduce the following data matrices:

$$
\begin{aligned}
\Delta\mathbf{G} &= [\delta\boldsymbol{g}_0, \delta\boldsymbol{g}_1, \cdots, \delta\boldsymbol{g}_{T-1}] \in \mathbb{R}^{r \times T}, \Delta\bar{\mathbf{G}} = [\delta\boldsymbol{g}_1, \delta\boldsymbol{g}_2, \cdots, \delta\boldsymbol{g}_T] \in \mathbb{R}^{r \times T}, \\
\mathbf{G}_x &= [\boldsymbol{g}(\boldsymbol{x}_0, \boldsymbol{\theta}), \boldsymbol{g}(\boldsymbol{x}_1, \boldsymbol{\theta}), \cdots, \boldsymbol{g}(\boldsymbol{x}_{T-1}, \boldsymbol{\theta})] \in \mathbb{R}^{r \times T}, \\
\bar{\mathbf{G}}_x &= [\boldsymbol{g}(\boldsymbol{x}_1, \boldsymbol{\theta}), \boldsymbol{g}(\boldsymbol{x}_2, \boldsymbol{\theta}), \cdots, \boldsymbol{g}(\boldsymbol{x}_T, \boldsymbol{\theta})] \in \mathbb{R}^{r \times T}, \\
\mathbf{X} &= [\boldsymbol{x}_0, \boldsymbol{x}_1, \cdots, \boldsymbol{x}_{T-1}] \in \mathbb{R}^{n \times T}, \bar{\mathbf{X}} = [\boldsymbol{x}_1, \boldsymbol{x}_2, \cdots, \boldsymbol{x}_T] \in \mathbb{R}^{n \times T}, \\
\mathbf{W} &= [\boldsymbol{w}_0, \boldsymbol{w}_1, \cdots, \boldsymbol{w}_{T-1}] \in \mathbb{R}^{n \times T}, \bar{\mathbf{W}} = [\boldsymbol{w}_1, \boldsymbol{w}_2, \cdots, \boldsymbol{w}_T] \in \mathbb{R}^{n \times T},
\end{aligned}
\tag{23}
$$

For brevity, we omit the iteration index $k$, as the constant matrices of $\tilde{\boldsymbol{f}}$ and $\bar{\boldsymbol{f}}$ are assumed to be computed using the same function $\boldsymbol{g}(\cdot, \boldsymbol{\theta})$. Utilizing Eq. (16) and Eq. (23), we obtain:

$$
\mathbf{Y} = \mathbf{X} + \mathbf{W}, \quad \bar{\mathbf{Y}} = \bar{\mathbf{X}} + \bar{\mathbf{W}}, \quad \mathbf{G} = \mathbf{G}_x + \Delta\mathbf{G}, \quad \bar{\mathbf{G}} = \bar{\mathbf{G}}_x + \Delta\bar{\mathbf{G}}.
\tag{24}
$$

We now proceed by minimizing Eq. (11) (over the noise-free trajectory) with respect to the dynamics matrices. The solution to this problem is analogous to the one derived in Eq. (18) (over the noisy trajectory). By utilizing the notations introduced in Eq. (23), the following results can be obtained through a reformulation of Eq. (18):

$$
[\tilde{\boldsymbol{A}}^*, \tilde{\boldsymbol{B}}^*] = \bar{\mathbf{G}}_x \begin{bmatrix} \mathbf{G}_x \\ \mathbf{U} \end{bmatrix}^{\dagger},
\tag{25}
$$

$$
\tilde{\boldsymbol{C}}^* = \mathbf{X}\mathbf{G}_x^{\dagger}.
\tag{26}
$$

We then expand Eq. (25)-(26) using Eq. (24) and the definition of the Moore–Penrose inverse. This results in the following expression:

$$
\begin{aligned}
&[\tilde{\boldsymbol{A}}^*, \tilde{\boldsymbol{B}}^*] \\
&= \bar{\mathbf{G}}_x \begin{bmatrix} \mathbf{G}_x \\ \mathbf{U} \end{bmatrix}^\dagger = (\bar{\mathbf{G}} - \Delta\bar{\mathbf{G}}) \begin{bmatrix} \mathbf{G} - \Delta\mathbf{G} \\ \mathbf{U} \end{bmatrix}^\dagger \\
&= (\bar{\mathbf{G}} - \Delta\bar{\mathbf{G}}) \begin{bmatrix} \mathbf{G} - \Delta\mathbf{G} \\ \mathbf{U} \end{bmatrix}' (\begin{bmatrix} \mathbf{G} - \Delta\mathbf{G} \\ \mathbf{U} \end{bmatrix} \begin{bmatrix} \mathbf{G} - \Delta\mathbf{G} \\ \mathbf{U} \end{bmatrix}')^{-1} \\
&= (\bar{\mathbf{G}} \begin{bmatrix} \mathbf{G} \\ \mathbf{U} \end{bmatrix}' - \underbrace{[(\bar{\mathbf{G}} - \Delta\bar{\mathbf{G}})\Delta\mathbf{G}' + \Delta\bar{\mathbf{G}}\mathbf{G}', \Delta\bar{\mathbf{G}}\mathbf{U}']}_{\mathbf{N}_w})(\underbrace{\begin{bmatrix} \mathbf{G} \\ \mathbf{U} \end{bmatrix} \begin{bmatrix} \mathbf{G} \\ \mathbf{U} \end{bmatrix}'}_{\mathbf{P}} + \underbrace{\begin{bmatrix} (\Delta\mathbf{G} - \mathbf{G})\Delta\mathbf{G}' - \Delta\mathbf{G}\mathbf{G}' & -\Delta\mathbf{G}\mathbf{U}' \\ -\mathbf{U}\Delta\mathbf{G}' & 0 \end{bmatrix}}_{\mathbf{M}_w} \mathbf{I}_{r+m})^{-1}
\end{aligned}
$$
(27)

and

$$
\begin{aligned}
\tilde{\boldsymbol{C}}^* &= \mathbf{X}\mathbf{G}_x^\dagger = (\mathbf{Y} - \mathbf{W})(\mathbf{G} - \Delta\mathbf{G})^\dagger, \\
&= (\mathbf{Y} - \mathbf{W})(\mathbf{G} - \Delta\mathbf{G})'((\mathbf{G} - \Delta\mathbf{G})(\mathbf{G} - \Delta\mathbf{G})')^{-1}, \\
&= (\mathbf{Y}\mathbf{G}' - \underbrace{((\mathbf{Y} + \mathbf{W})\Delta\mathbf{G}' - \mathbf{W}\mathbf{G}')}_{\bar{\mathbf{N}}_w})(\underbrace{\mathbf{G}\mathbf{G}'}_{\bar{\mathbf{P}}} + \underbrace{((-\mathbf{G} + \Delta\mathbf{G})\Delta\mathbf{G}' - \Delta\mathbf{G}\mathbf{G}')}_{\bar{\mathbf{M}}_w}\mathbf{I}_r)^{-1}.
\end{aligned}
$$
(28)

Note here that $[\bar{\boldsymbol{A}}^*, \bar{\boldsymbol{B}}^*] = \bar{\mathbf{G}}\begin{bmatrix} \mathbf{G} \\ \mathbf{U} \end{bmatrix}'(\begin{bmatrix} \mathbf{G} \\ \mathbf{U} \end{bmatrix}\begin{bmatrix} \mathbf{G} \\ \mathbf{U} \end{bmatrix}')^{-1}$ and $\bar{\boldsymbol{C}}^* = \mathbf{Y}\mathbf{G}'(\mathbf{G}\mathbf{G}')^{-1}$. Applying Sherman–Morrison formula to Eq. (27)-(28), that is, for given invertible matrix $P \in \mathbb{R}^{n \times n}$ and column vectors $m, v \in \mathbb{R}^n$, if $1 + v'P^{-1}m \neq 0$, the following holds:

$$
(P + mv')^{-1} = P^{-1} - \frac{P^{-1}mv'P^{-1}}{1 + v'P^{-1}m}.
$$

Using this formula, we derive the following results:

$$
[\tilde{\boldsymbol{A}}^*, \tilde{\boldsymbol{B}}^*] = [\bar{\boldsymbol{A}}^*, \bar{\boldsymbol{B}}^*] + (\mathbf{N}_w\mathbf{P}^{-1} - [\bar{\boldsymbol{A}}^*, \bar{\boldsymbol{B}}^*])\mathbf{M}_w\mathbf{P}^{-1}(\mathbf{I}_{r+m} + \mathbf{P}^{-1}\mathbf{M}_w)^{-1} - \mathbf{N}_w\mathbf{P}^{-1}
$$
(29)

and

$$
\tilde{\boldsymbol{C}}^* = \bar{\boldsymbol{C}}^* + (\bar{\mathbf{N}}_w\bar{\mathbf{P}}^{-1} - \bar{\boldsymbol{C}}^*)\bar{\mathbf{M}}_w\bar{\mathbf{P}}^{-1}(\mathbf{I}_r + \bar{\mathbf{P}}^{-1}\bar{\mathbf{M}}_w)^{-1} - \bar{\mathbf{N}}_w\bar{\mathbf{P}}^{-1}.
$$
(30)

Here, we recall the dynamics difference $\mathbf{L}_{f,2}(\bar{\boldsymbol{\theta}}^*)$ defined in Eq. (20), where we replace $\bar{\boldsymbol{\theta}}^*$ with $\boldsymbol{\theta}$, as $\bar{\boldsymbol{\theta}}^*$ represents the optimal solution of $\mathbf{L}_{f,1}$, which is found using the same variable $\boldsymbol{\theta}$ in the overall loss function $\mathbf{L}_f = \mathbf{L}_{f,1} + \mathbf{L}_{f,2}$ as defined in Eq. (14), given by:

$$
\mathbf{L}_{f,2}(\boldsymbol{\theta}) = \frac{1}{2T} \max_{\boldsymbol{w}_t}(\sum_{t=0}^{T-1} \|\boldsymbol{g}(\boldsymbol{x}_t, \tilde{\boldsymbol{\theta}}^*) - \boldsymbol{g}(\boldsymbol{y}_t, \boldsymbol{\theta})\|^2 + \|[\tilde{\boldsymbol{A}}^*, \tilde{\boldsymbol{B}}^*] - [\bar{\boldsymbol{A}}^*, \bar{\boldsymbol{B}}^*]\|_F^2 + \|\tilde{\boldsymbol{C}}^* - \bar{\boldsymbol{C}}^*\|_F^2).
$$

By following Eq. (29)-(30), $\mathbf{L}_{f,2}(\boldsymbol{\theta})$ becomes

$$
\begin{aligned}
\mathbf{L}_{f,2}(\boldsymbol{\theta}) = &\frac{1}{2T} \max_{\boldsymbol{w}_t}(\| (\mathbf{N}_w\mathbf{P}^{-1} - [\bar{\boldsymbol{A}}^*, \bar{\boldsymbol{B}}^*])\mathbf{M}_w\mathbf{P}^{-1}(\mathbf{I}_{r+m} + \mathbf{P}^{-1}\mathbf{M}_w)^{-1} - \mathbf{N}_w\mathbf{P}^{-1} \|_F^2 + \| (\bar{\mathbf{N}}_w\bar{\mathbf{P}}^{-1} - \bar{\boldsymbol{C}}^*) \\
&\bar{\mathbf{M}}_w\bar{\mathbf{P}}^{-1}(\mathbf{I}_r + \bar{\mathbf{P}}^{-1}\bar{\mathbf{M}}_w)^{-1} - \bar{\mathbf{N}}_w\bar{\mathbf{P}}^{-1} \|_F^2 + \sum_{t=0}^{T-1} \|\boldsymbol{g}(\boldsymbol{x}_t, \tilde{\boldsymbol{\theta}}^*) - \boldsymbol{g}(\boldsymbol{y}_t, \boldsymbol{\theta})\|^2).
\end{aligned}
$$
(31)

To proceed and for clarity, we define $\delta\boldsymbol{g}_t^{max} = \boldsymbol{g}(\boldsymbol{x}_t + \boldsymbol{w}_{max}, \boldsymbol{\theta}) - \boldsymbol{g}(\boldsymbol{x}_t, \boldsymbol{\theta})$ and

$$
\begin{aligned}
&\Delta\mathbf{G}_{max} = [\delta\boldsymbol{g}_0^{max}, \delta\boldsymbol{g}_1^{max}, \cdots, \delta\boldsymbol{g}_{T-1}^{max}] \in \mathbb{R}^{r \times T}, \Delta\bar{\mathbf{G}}_{max} = [\delta\boldsymbol{g}_1^{max}, \delta\boldsymbol{g}_2^{max}, \cdots, \delta\boldsymbol{g}_T^{max}] \in \mathbb{R}^{r \times T}, \\
&\mathbf{W}_{max} = [\boldsymbol{w}_{max}, \boldsymbol{w}_{max}, \cdots, \boldsymbol{w}_{max}] \in \mathbb{R}^{n \times T}, \mathbf{Y}_{max} = \mathbf{X} + \mathbf{W}_{max}.
\end{aligned}
$$

Accordingly, we define the following matrices under the maximum measurement noise:

$$
\begin{aligned}
\mathbf{N}_{max} &= [(\bar{\mathbf{G}} - \Delta\bar{\mathbf{G}}_{max})\Delta\mathbf{G}'_{max} + \Delta\bar{\mathbf{G}}_{max}\mathbf{G}', \Delta\bar{\mathbf{G}}_{max}\mathbf{U}'], \\
\bar{\mathbf{N}}_{max} &= (\mathbf{Y}_{max} + \mathbf{W}_{max})\Delta\mathbf{G}'_{max} - \mathbf{W}_{max}\mathbf{G}', \\
\mathbf{M}_{max} &= \begin{bmatrix} (\Delta\mathbf{G}_{max} - \mathbf{G})\Delta\mathbf{G}'_{max} - \Delta\mathbf{G}_{max}\mathbf{G}' & -\Delta\mathbf{G}_{max}\mathbf{U}' \\ -\mathbf{U}\Delta\mathbf{G}'_{max} & 0 \end{bmatrix}, \\
\bar{\mathbf{M}}_{max} &= (-\mathbf{G} + \Delta\mathbf{G}_{max})\Delta\mathbf{G}'_{max} - \Delta\mathbf{G}_{max}\mathbf{G}'.
\end{aligned}
\tag{32}
$$

Then, by applying the triangle inequality and utilizing the fact that $\boldsymbol{g}(\boldsymbol{x}, \boldsymbol{\theta})$ is Lipschitz continuous with Lipschitz constants $L_x$ and $L_\theta$, where $L_x$ denotes the Lipschitz constant of $\boldsymbol{g}$ with respect to $\boldsymbol{x}$ and $L_\theta$ denotes the Lipschitz constant of $\boldsymbol{g}$ with respect to $\boldsymbol{\theta}$, the function $\mathbf{L}_{f,2}(\boldsymbol{\theta})$ can be expressed as:

$$
\begin{aligned}
\mathbf{L}_{f,2}(\boldsymbol{\theta}) =& \frac{1}{2T}(\| (\mathbf{N}_{max}\mathbf{P}^{-1} - [\bar{\boldsymbol{A}}^*, \bar{\boldsymbol{B}}^*])\mathbf{M}_{max}\mathbf{P}^{-1}(\mathbf{I}_{r+m} + \mathbf{P}^{-1}\mathbf{M}_{max})^{-1} - \mathbf{N}_{max}\mathbf{P}^{-1} \|_F^2 \\
&+ \| (\bar{\mathbf{N}}_{max}\bar{\mathbf{P}}^{-1} - \bar{\boldsymbol{C}}^*)\bar{\mathbf{M}}_{max}\bar{\mathbf{P}}^{-1}(\mathbf{I}_r + \bar{\mathbf{P}}^{-1}\bar{\mathbf{M}}_{max})^{-1} - \bar{\mathbf{N}}_{max}\bar{\mathbf{P}}^{-1} \|_F^2 \\
&+ \sum_{t=0}^{T-1} \|\boldsymbol{g}(\boldsymbol{x}_t, \tilde{\boldsymbol{\theta}}^*) - \boldsymbol{g}(\boldsymbol{x}_t + \boldsymbol{w}_{max}, \boldsymbol{\theta})\|^2) \\
\leq& \frac{1}{2T}(\| \mathbf{N}_{max}\mathbf{P}^{-1} \|_F^2 + \| [\bar{\boldsymbol{A}}^*, \bar{\boldsymbol{B}}^*] \|_F^2) \| \mathbf{M}_{max}\mathbf{P}^{-1} \|_F^2 \| (\mathbf{I}_{r+m} + \mathbf{P}^{-1}\mathbf{M}_{max})^{-1} \|_F^2 + \| \mathbf{N}_{max}\mathbf{P}^{-1} \|_F^2 \\
&+ (\| \bar{\mathbf{N}}_{max}\bar{\mathbf{P}}^{-1} \|_F^2 + \| \bar{\boldsymbol{C}}^* \|_F^2) \| \bar{\mathbf{M}}_{max}\bar{\mathbf{P}}^{-1} \|_F^2 \| (\mathbf{I}_r + \bar{\mathbf{P}}^{-1}\bar{\mathbf{M}}_{max})^{-1} \|_F^2 + \| \bar{\mathbf{N}}_{max}\bar{\mathbf{P}}^{-1} \|_F^2 + TL_g^2),
\end{aligned}
\tag{33}
$$

where $L_g = \sqrt{2}\,max\{L_x, L_\theta\}$. Observing that the upper bound in Eq. (33) contains complex terms $\| (\mathbf{I}_{r+m} + \mathbf{P}^{-1}\mathbf{M}_{max})^{-1} \|_F^2$ and $\| (\mathbf{I}_r + \bar{\mathbf{P}}^{-1}\bar{\mathbf{M}}_{max})^{-1} \|_F^2$, which complicate the minimization of the upper bound by tuning $\boldsymbol{\theta}$, we propose an alternative approach. Instead of attempting to find a solution that makes $\| (\mathbf{I}_{r+m} + \mathbf{P}^{-1}\mathbf{M}_{max})^{-1} \|_F^2 = 0$ and $\| (\mathbf{I}_r + \bar{\mathbf{P}}^{-1}\bar{\mathbf{M}}_{max})^{-1} \|_F^2 = 0$, we aim to achieve $\| \mathbf{P}^{-1}\mathbf{M}_{max} \|_F^2 = 0$ and $\| \bar{\mathbf{P}}^{-1}\bar{\mathbf{M}}_{max} \|_F^2 = 0$ such that $\| (\mathbf{I}_{r+m} + \mathbf{P}^{-1}\mathbf{M}_{max})^{-1} \|_F^2 = r + m$ and $\| (\mathbf{I}_r + \bar{\mathbf{P}}^{-1}\bar{\mathbf{M}}_{max})^{-1} \|_F^2 = r$, and then proceed to minimize the remaining terms. Thus, we define the upper bound of Eq. (33) as:

$$
\begin{aligned}
\tilde{\mathbf{L}}_{f,2}(\boldsymbol{\theta}) =& \frac{1}{2T}\Big( \| \mathbf{N}_{max}\mathbf{P}^{-1} \|_F^2 + \| [\bar{\boldsymbol{A}}^*, \bar{\boldsymbol{B}}^*] \|_F^2) \| \mathbf{M}_{max}\mathbf{P}^{-1} \|_F^2 \| \mathbf{P}^{-1}\mathbf{M}_{max} \|_F^2 + \| \mathbf{N}_{max}\mathbf{P}^{-1} \|_F^2 \\
&+ (\| \bar{\mathbf{N}}_{max}\bar{\mathbf{P}}^{-1} \|_F^2 + \| \bar{\boldsymbol{C}}^* \|_F^2) \| \bar{\mathbf{M}}_{max}\bar{\mathbf{P}}^{-1} \|_F^2 \| \bar{\mathbf{P}}^{-1}\bar{\mathbf{M}}_{max} \|_F^2 + \| \bar{\mathbf{N}}_{max}\bar{\mathbf{P}}^{-1} \|_F^2 + TL_g^2) \\
\leq& \frac{1}{2T}\Big( \| \mathbf{N}_{max} \|_F^2 \| \mathbf{P}^{-1} \|_F^2 + \| [\bar{\boldsymbol{A}}^*, \bar{\boldsymbol{B}}^*] \|_F^2) \| \mathbf{M}_{max} \|_F^4 \| \mathbf{P}^{-1} \|_F^4 + \| \mathbf{N}_{max} \|_F^2 \| \mathbf{P}^{-1} \|_F^2 \\
&+ (\| \bar{\mathbf{N}}_{max} \|_F^2 \| \bar{\mathbf{P}}^{-1} \|_F^2 + \| \bar{\boldsymbol{C}}^* \|_F^2) \| \bar{\mathbf{M}}_{max} \|_F^2 \| \bar{\mathbf{P}}^{-1} \|_F^4 + \| \bar{\mathbf{N}}_{max} \|_F^2 \| \bar{\mathbf{P}}^{-1} \|_F^2 + TL_g^2).
\end{aligned}
\tag{34}
$$

Here, using the the Lipschitz continuity of $\boldsymbol{g}$ and Eq. (32), one obtains:

$$
\begin{aligned}
\| \mathbf{N}_{max} \|_F^2 &\leq (\| \bar{\mathbf{G}} \|_F^2 + \| \mathbf{G} \|_F^2 + (TL_x w_{max})^2 + \| \mathbf{U} \|_F^2)(TL_x w_{max})^2, \\
\| \mathbf{M}_{max} \|_F^2 &\leq (2 \| \mathbf{G} \|_F^2 + 2 \| \mathbf{U} \|_F^2 + (TL_x w_{max})^2)(TL_x w_{max})^2, \\
\| \bar{\mathbf{N}}_{max} \|_F^2 &\leq (\| \mathbf{Y} \|_F^2 + (Tw_{max})^2)(TL_x w_{max})^2 + (Tw_{max})^2 \| \mathbf{G} \|_F^2, \\
\| \bar{\mathbf{M}}_{max} \|_F^2 &\leq (2 \| \mathbf{G} \|_F^2 + (TL_x w_{max})^2)(TL_x w_{max})^2.
\end{aligned}
\tag{35}
$$

Finally, using Eq. (35) and removing the constant terms while merging the repeated terms from Eq. (34), the resulting loss function is expressed as:

$$
\hat{\mathbf{L}}_{f,2}(\boldsymbol{\theta}) = \frac{1}{2T}\Big( \| \mathbf{P}^{-1} \|_F^2 ((\| \bar{\mathbf{G}} \|_F^2 + \| [\bar{\boldsymbol{A}}^*, \bar{\boldsymbol{B}}^*] \|_F^2) \| \mathbf{G} \|_F^2 + \| \bar{\mathbf{G}} \|_F^2) + \| \bar{\mathbf{P}}^{-1} \|_F^2 \| \bar{\boldsymbol{C}}^* \|_F^2 \| \mathbf{G} \|_F^2 \Big),
\tag{36}
$$

where $[\bar{\boldsymbol{A}}^*, \bar{\boldsymbol{B}}^*] = \bar{\mathbf{G}} \begin{bmatrix} \mathbf{G} \\ \mathbf{U} \end{bmatrix}^\dagger$ and $\bar{\boldsymbol{C}}^* = \mathbf{Y}\mathbf{G}^\dagger$. ∎

## A.2   Simulation details

In this subsection, we provide the simulation details regarding the experiment in Section 4.

### A.2.1   Computation resource and training parameters

|  | 2D dynamics | Cartpole | Lunar lander | Surface vehicle |
|---|---|---|---|---|
| Optimizer | Adam | | | |
| Accuracy ($\epsilon$) | $1e-4$ | | | |
| Training epochs ($S$) | 1e4 | | | |
| Learning rate ($\alpha_k$) | $1e-5$ | | | |
| The number of data pairs (T) | 500 | 600 | 1600 | 600 |
| Compute device | Apple M2, 16GB RAM | | | |

Table 3: Training parameters.

### A.2.2   DNNs architecture

The DNN architectures of method DKND and DK used in this paper are presented in Table 4. We refer to `https://pytorch.org/docs/stable/nn.html` for the definition of functions $Linear()$, $ReLU()$ and we denote layer$^i$ as the $i$-th layer of the DNN and $Linear([n, m])$ denotes a linear function with a weight matrix of shape $n \times m$. Since for the DKND and DK methods, the input of its DNN observable function is the

|  | 2D dynamics | Cartpole | Lunar lander | Surface vehicle |
|---|---|---|---|---|
| layer$^1$ type | $Linear([2, 512])$ | $Linear([4, 512])$ | $Linear([6, 512])$ | $Linear([6, 512])$ |
| layer$^2$ type | $ReLU()$ | $ReLU()$ | $ReLU()$ | $ReLU()$ |
| layer$^3$ type | $Linear([512, 128])$ | $Linear([512, 128])$ | $Linear([512, 128])$ | $Linear([512, 128])$ |
| layer$^4$ type | $ReLU()$ | $ReLU()$ | $ReLU()$ | $ReLU()$ |
| layer$^5$ type | $Linear([128, 4])$ | $Linear([128, 6])$ | $Linear([128, 4])$ | $Linear([128, 10])$ |

Table 4:   DNN structures of DKND and DK.

measured state $\boldsymbol{y}_t$ and the input of the MLP method is a stacked vector of $[\boldsymbol{y}_t', \boldsymbol{u}_t']'$, we show the DNN structure of the MLP method in the following table.

|  | 2D dynamics | Cartpole | Lunar lander | Surface vehicle |
|---|---|---|---|---|
| layer$^1$ type | $Linear([3, 512])$ | $Linear([5, 512])$ | $Linear([8, 512])$ | $Linear([8, 512])$ |
| layer$^2$ type | $ReLU()$ | $ReLU()$ | $ReLU()$ | $ReLU()$ |
| layer$^3$ type | $Linear([512, 128])$ | $Linear([512, 128])$ | $Linear([512, 128])$ | $Linear([512, 128])$ |
| layer$^4$ type | $ReLU()$ | $ReLU()$ | $ReLU()$ | $ReLU()$ |
| layer$^5$ type | $Linear([128, 4])$ | $Linear([128, 6])$ | $Linear([128, 4])$ | $Linear([128, 10])$ |

Table 5:   DNN structures of MLP.

