# OpenReview forum: "Deep Koopman Learning using Noisy Data"
_TMLR — Accepted by TMLR_

### Review · Reviewer_SPcz · 2025-02-13

**Summary Of Contributions:**

A method for learning observables of Koopman operators with observation noise is proposed. The idea is based on the upper-bounding of noise effects, resulting in a kind of regularization terms in the final objective function. The performance of the method is examined with four types of simulated data.

**Audience:**

Yes

**Claims And Evidence:**

No

**Requested Changes:**

(1) Most importantly, please elaborate on the proof of the main theorem, in particular around the transition from Eq. (33) to Eq. (35).

(2) Additionally but still importantly, please consider adding experiments on real-world data.

**Strengths And Weaknesses:**

### (0)
The problem, observable learning on noisy data, is challenging but important. The paper concisely motivates it and discuss relevant previous studies.

### (1)
I have two major concerns. One is about the theoretical construction of the method. In my understanding, the core of the method is the transition from Eq. (33) to Eq. (35), which claims that the minimizer of (35) coincides with that of (33). I cannot follow the reasoning here. Let us think of the following two minimization problems:
- $\operatorname{minimize}_\theta ~ f(\theta) + || ( h(\theta) - g(\theta) ) c(\theta) - h(\theta)  ||^2$ ... (A)
- $\operatorname{minimize}_\theta ~ f(\theta) + || g(\theta) ||^2 + || h(\theta) - g(\theta) ||^2 $ ... (B)

I think they are analogous to (33) and (35) (before the inequality); with $f(\theta) = || \delta g_k ||^2$, $g(\theta) = [ \bar{A}^*, \bar{B}^* ]$, and $h(\theta) = n v' p^{-1}$, $c(\theta) = u v' p^{-1} (I + v' p^{-1} u)^{-1}$.
However, in general, (A) and (B) do not have the same minimizers, a minimal counterexample being $f(\theta)=\theta^2$, $h(\theta)=\theta$, $g(\theta)=c(\theta)=1$.
Moreover, taking the upper bound of the first-second lines of (35), the minimizer is not necessarily preserved, either.
Thus the reasoning around (33) and (35) seems to need further clarification.

### (2)
Another concern is about the empirical evaluation. Although the method is motivated by noisy measurements, no experiments are conducted on any real-world data. This may not necessarily be a fatal drawback resulting in an immediate rejection but certainly diminishes the convincingness of the paper.

### (3)
Here are other minor points:
- Page 1: "the Koopman operator fails ..." --> what do you mean by "fails"?
- Page 1-2: "dynamic mode decomposition (DMD) ... lift the state space to a higher-dimensional space" --> the standard DMD does not do the lifting.
- Page 5, in Remark 1: "invertible" --> right-invertible
- Page 5, just after (24): "$w_1 \geq 0$, ... $w_6 \geq 0$" --> from where these weights appeared? No explanation found.
- The Gaussian and Poisson noises are not bounded.

2025-02-14: a small typo corrected

---

> ### Author Response · Authors · 2025-02-13
> **a simple follow-up to clarify your question.**
>
> Thank you for your quick response and your valuable suggestions!
>
> I just want to clarify your first main concern before I start addressing your questions and comments.
>
> In your example, i.e., minimization (B), you mean $f(\theta) + \parallel h(\theta) \parallel^2 + \parallel h(\theta) - g(\theta) \parallel^2$, do I understand it correctly?
>
> Best,

---

> ### Comment · Reviewer_SPcz · 2025-02-13
>
> Thanks for checking the review. As for your question, no I did; I did mean $g$, not $h$, in the second term. This should correspond to the transition from Eq. (33) to Eqs. (34) and (35), where the $nv'p^{-1}$ became $[\bar{A}^*, \bar{B}^*]$.

---

> ### Author Response · Authors · 2025-03-04
> **Author response**
>
> We sincerely appreciate your time and valuable feedback. In response to your comments, we have made the following key revisions, which are highlighted in blue in the attached revised draft:
>
> 1.	As suggested, we have revised the proof section, particularly the transition between Equations (33–35). The updated proof is now more straightforward, as we focus on reformulating the optimization problem by solely identifying the upper bound and eliminating constant terms.
>
> 2.	We have updated the surface vehicle example to utilize data from real surface vehicle robots. The details of this revision can be found in the Simulation Section of the revised manuscript, marked in blue.
>
> Author response to your concerns and questions:
>
> 1.	Regarding your questions on Optimization Problems A and B, these two problems are not equivalent. In our initial draft, our reasoning was based on the assumption that if we achieve $h(\theta) = g(\theta)$, then Eq. (33) can be simplified. However, before reaching this condition, the two problems do not share the same minimizer. In the revised draft, we have restructured Eqs. (33)–(35) by focusing solely on deriving an upper bound to reformulate the minimization problem. Additionally, the minimizer you suggested as an example may not exist because in the first draft $c(θ)=uv′p−1(I+v′p−1u)^{-1}$, and there is no solution for matrices $AB -BA = I$, refer to Artin's Algebra.
> 2.	In the revised draft, we have incorporated real-world data for the simulation. The reason this was not included in the initial draft is that our problem formulation focuses on achieving a good mapping approximation between $y_t$, $u_t$ and $x_{t+1}$. However, real-world data typically contains unknown noise, making it difficult to determine the true x values. This uncertainty complicates the evaluation of the true performance of our approach.
> 3.	We apologize for the confusion regarding the noisy bound in the simulation section. Our intent was not to suggest that the noise distribution is inherently bounded. Rather, we meant that the noise is clipped when generating noisy data. We have clarified this point in the revised draft.

---

> > ### Comment · Reviewer_SPcz · 2025-03-26
> >
> > I am sorry for my late reply. Thank you very much for the clarification.
> >
> > The new experimental result is nice. My intention of suggesting adding real-world experiments was not necessarily about conducting strictly quantitative evaluation but rather to show the applicability of the proposed method to real-world data. The new experiment seems to be fine in this regard.
> >
> > As for the new direction of the theoretical construction, the reformulation focusing on the identification of an upper bound sounds reasonable. What I am still not fully convinced is the following claim remaining there:
> >
> > > Note here that the minimization of Eq. (38) is not equal to the minimization of Eq. (33) but they share the same optimal solution.
> >
> > For saying "they share the same optimal solution," I think a few more arguments are needed; why the two problems share the same optimal solution is still not clear.
> >
> > With that being said, I do not think such a claim is necessary --- as suggested by the authors, simply saying the method minimizes an upper bound of the original loss would totally suffice, and the consistency of the minimizers does not seem so important.
> >
> > Anyway this is not something I seriously require for acceptance. Please just consider further polishing the description.

---

### Review · Reviewer_MLZK · 2025-02-13

**Summary Of Contributions:**

This paper studies the learning of a nonlinear dynamical system under noisy measurements of the states. Driven by the Koopman operator theory, it proposes to linearize the dynamical system by nonlinearly embedding the state vector into a higher-dimensional observation space. Building upon the linear system, the paper derives a DKND algorithm to handle the noises and implements it by alternatively optimizing the measurement embedding and the linear system. Numerical experiments are shown.

**Audience:**

Yes

**Claims And Evidence:**

Yes

**Requested Changes:**

There are a few comments on the presentation of the work:
1. The most crucial comment is that I would like to see more content on how the proposed method handles the noises and why it is better than other methods, as this is the core contribution of the paper. So far, the majority of the content in section 3 is spent on deriving the algorithm. While this is important, perhaps the derivation can be compressed for more discussion about the clear properties of the proposed algorithm.

2. The figures on page 8-10 certainly need reorganization. They should be made more compact so that a page can fit at least two of them. Captures should be expanded to explain what they are. If possible, the boxes of the four subfigures (a)-(d) should be aligned.

3. On page 3, you wrote "$g(\cdot, \boldsymbol{\theta})$ is Lipschitz continuous with Lipschitz constant $L_g$." The authors should make it clear which variable is changing. If you mean that $g$ is Lipschitz continuous in the first variable fixed a fixed $\boldsymbol{\theta}$, does $L_g$ depend on $\boldsymbol{\theta}$? Finally, I don't see $L_g$ pop up again later in the manuscript. Did I miss something, or why is $L_g$ important?

4. There are also many minor typos. I suggest that the authors proofread the manuscript again before resubmitting:

   1. In the last paragraph of page 2, $\xi$ should be made bold. Also, what is $\hat{\mathbf{u}}\_t$?
   2. In (5) and (6), it doesn't make sense that $\hat{\mathbf{x}}\_{t+1} = \mathbf{C} g(\hat{\mathbf{x}}\_{k+1}, \boldsymbol{\theta})$.
   3. After (6), the expression $g(\cdot, \boldsymbol{\theta}): \mathbb{R}^n \times \mathbb{R}^p \rightarrow \mathbb{R}^r$ should be changed to either $g(\cdot, \cdot): \mathbb{R}^n \times \mathbb{R}^p \rightarrow \mathbb{R}^r$ or $g(\cdot, \boldsymbol{\theta}): \mathbb{R}^n \rightarrow \mathbb{R}^r$.
   4. In the second equation of the "Key ideas" subsection, $\tilde{\mathbf{C}}$ should be changed into $\overline{\mathbf{C}}$.
   5. At the end of page 4, (20) is an objective instead of a problem. You should not say "there exists a solution to (20)."
   6. I don't know the precise formatting rules of TMLR, but I recommend using parentheses for equations (e.g., (2) instead of 2) unless the formatting rules suggest otherwise.

**Strengths And Weaknesses:**

Strengths:
   * The paper studies an important problem where noises are presented. The derivations presented in the paper are rigorous and the statements are mathematically correct.

Weaknesses:
   * While the paper presents some theorems and derivations to motivate the final algorithm, the algorithm itself lacks theoretical guarantees. It would be interesting to analyze the algorithm more and understand how it compares to other existing methods. Convergence results, if any, would also be helpful.
   * The presentation of the paper can be improved. See comments below.
   * The empirical advantages of the proposed DKND algorithm are not clear from the numerical simulations. It seems that the gain is only marginal.

---

> ### Author Response · Authors · 2025-03-04
> **Author response**
>
> We appreciate the reviewer’s time and valuable comments and suggestions. In response, we have made the following key revisions, which are highlighted in blue in the latest uploaded draft:
>
> 1.	As suggested, we have refined the Challenges and Key Ideas section to focus solely on the key ideas of our work. Additionally, we have introduced Remark 1 to further clarify the fundamental mechanism of our approach. These changes are clearly marked in the draft.
>
> 2.	We have improved the algorithm, re-executed the simulations, and reorganized the plots for better clarity and coherence.
>
> 3.	To streamline notation, we have removed unnecessary symbols. For instance, we have moved the Lipschitz constant from the main text to the appendix, where it is explicitly defined when necessary. Similar refinements have been applied to other notations.
> 4.	We have addressed minor comments and revised the manuscript to align with the standards suggested by the reviewer.
>
> Author response to your concerns and questions:
>
> 1.	We apologize for the confusion regarding the equations. After reviewing your comments, we realized that the TMLR template does not automatically enclose equations in brackets. We have addressed this issue in the revised draft and have also checked for similar inconsistencies throughout the manuscript.
>
> 2.	Thank you for your comments. The estimation errors across all gradient-based methods remain marginal over the training data, as they are terminated based on the same terminal accuracy criterion. We have explicitly highlighted this point in the revised draft, with the changes marked in blue.
>
> 3.	Our assumption of a linear mapping between $g$ and $x$ was made to simplify the controller design. However, we acknowledge that this assumption may not always be reasonable, as the $g$ function can be highly complex. We are aware that some approaches employ a decoder DNN to model this mapping. In our experiments with the inverted pendulum simulation, we attempted to use a decoder but training it to map $g$ to $x$ proved challenging. Instead, we found that using the $C$ matrix provided a more effective and reliable solution.

---

### Review · Reviewer_Ch1T · 2025-02-21

**Summary Of Contributions:**

The paper addresses the problem of learning a finite-dimensional approximation of the Koopman operator from system data that are corrupted by unknown but bounded measurement noise.

The main motivation of their work is that traditional data-driven Koopman methods typically assume either noise-free measurements or require strong noise assumptions, limiting their applicability in real-world settings. In this paper, these authors aim to relax these assumptions by only requiring that the noise be bounded.

In addition to learning the governing matrices A,B of the linear system, they also learn an additional matrix C which maps the lifted state and the original state.

**Audience:**

Yes

**Claims And Evidence:**

Yes

**Requested Changes:**

1) As seen from (3), they assume a full state feedback which is much of a simplification to the standard measurement mode y=Hx+w. This weakens the contribution and applicability of their to partial observable systems.

2) Why is it assumed that the relation between the lifted state g and the original state xhat is linear?

3) The advantage of their approach is only when they assume that the measurement noise is bounded (Theorem 1). If that is the case, how is your approach different from linear systems with bounded additive external disturbances acting on the systems. For these type of system, the research literature is quite mature.

4) There is a lot of previous work in literature that have considered stochastic systems with different noise characteristics [1-5]. How is our approach different from these, especially in the problem they are considering?

**Strengths And Weaknesses:**

The strengths of the paper are as follows:

1) The authors build on the idea of the deep Koopman operator, which relies on finding a suitable “lifted” space of observables (via a deep neural network) in which the system behaves linearly.

2) Unlike standard deep Koopman approaches that directly fit noisy input-output data, the paper introduces an additional term in the loss function to account for bounded noise and mitigate its impact in the learned Koopman operator.

3) One of the key sights of their approach is that after states are transformed by nonlinear observables, the noise is no longer simply additive or uncorrelated.

4) Three different noise distributions (Gaussian, Poisson, Uniform) were tested with various magnitudes of noise, demonstrating that their approach (which they termed as DKND) consistently achieves more accurate predictions (in terms of lower RMS error) compared to baseline methods (DMDTLS, DKL, MLP).

The weakness of the paper are as follows:

1) As seen from (3), they assume a full state feedback which is much of a simplification to the standard measurement mode y=Hx+w. This weakens the contribution and applicability of their to partial observable systems.

2) Why is it assumed that the relation between the lifted state g and the original state xhat is linear?

3) The advantage of their approach is only when they assume that the measurement noise is bounded (Theorem 1). If that is the case, how is your approach different from linear systems with bounded additive external disturbances acting on the systems. For these type of system, the research literature is quite mature.

4) There is a lot of previous work in literature that have considered stochastic systems with different noise characteristics [1-5]. How is our approach different from these, especially in the problem they are considering?



[1] Wanner, Mathias, and Igor Mezic. "Robust approximation of the stochastic Koopman operator." SIAM Journal on Applied Dynamical Systems 21, no. 3 (2022): 1930-1951.

[2] Sinha, Subhrajit, Bowen Huang, and Umesh Vaidya. "On robust computation of koopman operator and prediction in random dynamical systems." Journal of Nonlinear Science 30, no. 5 (2020): 2057-2090.

[3] Mauroy, Alexandre, and Jorge Goncalves. "Linear identification of nonlinear systems: A lifting technique based on the Koopman operator." In 2016 IEEE 55th Conference on Decision and Control (CDC), pp. 6500-6505. IEEE, 2016.

[4] Sinha, Subhrajit, Bowen Huang, and Umesh Vaidya. "Robust approximation of koopman operator and prediction in random dynamical systems." In 2018 Annual American Control Conference (ACC), pp. 5491-5496. IEEE, 2018.

[5] Haseli, Masih, and Jorge Cortés. "Approximating the Koopman operator using noisy data: noise-resilient extended dynamic mode decomposition." In 2019 American Control Conference (ACC), pp. 5499-5504. IEEE, 2019.

---

> ### Author Response · Authors · 2025-03-04
> **Author responses**
>
> Thank you for your time and for providing valuable comments and suggestions. We appreciate your thorough review, and we have carefully addressed your concerns. Below, we outline the main revisions made to our manuscript, which can be found in the revised draft via the latest uploaded draft. The main changes are highlighted in blue for clarity:
>
> 1.	Based on your suggestions, we have added the recommended references in the latest draft. We would also like to clarify that reference [5] was already included in our first draft, and we also have the reference of robust Koopman field of ‘Sinha et al. (2023)’ in the introduction section in our first draft.
>
> 2.	We added relevant remarks in the paper draft to clarify the potential confusion.
>
> Author response to your concerns and questions:
>
> 1.	Thank you for pointing it out for the measurement definition of this paper. We made this assumption of $y = x+w$ instead of $y = Hx + w$ because: a.	If partial observations were considered, additional rigorous results on observability would be required, especially for why we can map the lifting partial observations back to the original space. We recognize this as an important direction for future work. b.	We believe $y = x + w$ is a common problem setup, as reflected in the suggested references [1]-[5], all of which adopt this measurement $y = x + w$. c.	Under full observation with a known $H$, the resulting formulation would not significantly enhance the contribution, as the problem setup would remain similar.
>
> 2.	We assume the linear mapping between $g$ and $x$ is because: a.	 We assume a linear mapping to ultimately simplify the controller design. However, we acknowledge that this assumption may not always be reasonable, as g could be a complex function. b.	While some approaches employ a decoder DNN to represent this mapping, our experiments (e.g., in the inverted pendulum simulation) revealed that training a decoder to map $g$ to $x$ is challenging. Instead, using a linear C matrix proved to be a more practical approach according to our experiment for the pendulum example.
>
> 3.	Sorry for the confusion, this work mainly focuses on finding a good mapping between $y_t$, $u_t$, and $x_{t+1}$, with a key focus on addressing the distortion of original noise introduced by the Koopman basis function and how to mitigate this issue. While many existing studies on linear systems with bounded disturbances assume partially known dynamics (e.g., known dynamic matrices), our work considers a scenario where the system dynamics are completely unknown.
>
> 4.	Thank you for your comments. The problems in the literature on robust Koopman learning as you mentioned are mainly about learning the Koopman operator using noisy data, and the problem this work wants to solve is learning the Koopman operator from the noisy data plus learning the mapping between the Koopman basis function to the original space under the noisy data. Methodologically, our approach is innovative in that it reformulates the discrepancy between the Koopman operator learned from noisy versus noise-free data as a matrix norm term. This term is then minimized by optimizing the parameters of the parameterized Koopman basis function.

---

### Decision · Action_Editor_7jLu · 2025-03-27

**Recommendation:** Accept as is

**Comment:**

2 of the 3 reviewers recommended acceptance. The third believed that the paper was weak in its impact, and therefore recommended reject (although this is not grounds for rejection in the TMLR case). I agree that this work could be stronger, but I believe it is of sufficient interest as is to the broader community. Therefore, I am recommending accept as is.

**Audience:**

All 3 reviewers agree that this submission is of interest to the TMLR community. I agree, and believe that therefore this criteria for TMLR acceptance is satisfied.

**Claims And Evidence:**

All 3 reviewers agree that the claims were sufficiently supported by accurate and clear evidence. Therefore, this criteria for TMLR acceptance is satisfied.